# META-LEARNING RUNGE-KUTTA

## ABSTRACT

Initial value problems, *i.e.* differential equations with specific, initial conditions, represent a classic problem within the field of ordinary differential equations (ODEs). While the simplest types of ODEs may have closed-form solutions, most interesting cases typically rely on iterative schemes for numerical integration such as the family of Runge-Kutta methods. They are, however, sensitive to the strategy the step size is adapted during integration, which has to be chosen by the experimenter. Here, we show how the design of a step size controller can be cast as a learning problem, allowing deep networks to learn to exploit structure in the initial value problem at hand in an automatic way. The key ingredients for the resulting Meta-Learning Runge-Kutta (MLRK) are the development of a good performance measure and the identification of suitable input features. Traditional approaches suggest the local error estimates as input to the controller. However, by studying the characteristics of the local error function we show that including the partial derivatives of the initial value problem is favorable. Our experiments demonstrate considerable benefits over traditional approaches. In particular, MLRK is able to mitigate sudden spikes in the local error function by a faster adaptation of the step size. More importantly, the additional information in the form of partial derivatives and function values leads to a substantial improvement in performance. The source code can be found at `https://www.dropbox.com/sh/rkctdfhkosywnnx/AABKadysCR8-aHW_0kb6vCtSa?dl=0`

## 1 INTRODUCTION

Differential equations in their general form cover an extremely wide variety of disciplines: While many applications are rather intuitive, as for instance simple Newtonian physics and engineering, other more exotic use cases include the governing of price evolution in economics (Black & Scholes, 1973), the study of rates in chemical reactions (Scholz & Scholz, 2014), and the modeling of population growths in biology (Lotka, 1925; Volterra, 1926). In medicine, differential equations may be used to model cancer growth (Ilea et al., 2013), diabetes and the glucose metabolism (Esna-Ashari et al., 2017) as well as for pharmaceutical drug design (Deuflhard, 2000). Recently, differential equations have also been used as a way to design neural networks (Chen et al., 2018). Unfortunately, finding an analytical solution in closed form is in many cases very difficult, if not impossible. Therefore, a variety of numerical integration methods have been developed to obtain accurate, but approximate solutions. Arguably, the most prominent ones are Runge-Kutta methods, a family of integration methods for initial value problems. However, setting up Runge-Kutta involves several design choices, one of which is the step size controller. Using an adaptive step size strategy instead of a constant step size can often increase efficiency by several orders of magnitude, *c.f.* (Söderlind, 2006). Their performance is hampered by the fact that they only make use of hand-designed features.

**Contribution.** We show how to cast the design of a step size controller for Runge-Kutta as a learning problem, *c.f.* Fig. 1. The key ingredients of the resulting Meta-Learning Runge-Kutta (MLRK) are the identification of a good performance measure and appropriate inputs.

**Related Work.** Various approaches to control the step sizes in Runge-Kutta methods have been proposed. While some rely on signal processing principles where the goal is to produce a smooth step size sequence in conjunction with acceptable local errors, others are based on the assumption that step sizes should be adapted to a prescribed function of the solution, *e.g.* to preserve structure in geometric integration (Söderlind, 2006). In this work we will focus on control theoretic approaches which aim to keep the local error associated with a single step close to a tolerance parameter.

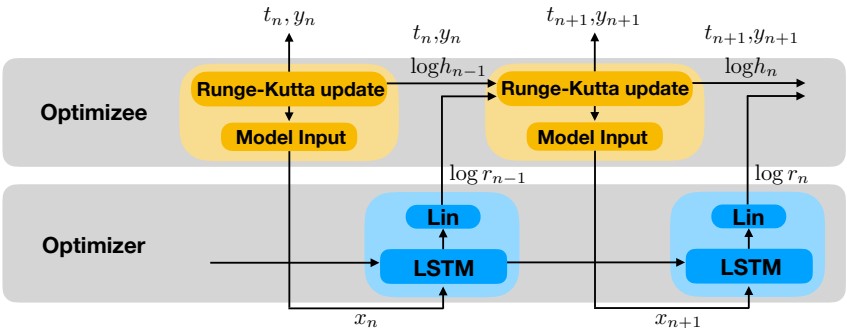

Figure 1: Meta-Learning Runge-Kutta: The model represented by the blue block determines the step size adjustment $\log r_{n-1}$ using a LSTM and a linear layer, *c.f.* Eq. (10). The Runge-Kutta update then determines the new step size $\log h_n = \log h_{n-1} + \log r_{n-1}$. The new step size is used to perform the next Runge-Kutta step which computes the approximation $y_{n+1}$ at time step $t_{n+1}$ and an error estimate $\text{err}_{n+1}$ according to (12). Then the next input $x_{n+1}$ for the model is computed.

Specifically, the design of the step size control algorithm is learned within MLRK. Indeed, casting algorithm design as a learning problem is not new. Andrychowicz et al. (2016) learned a gradient-based optimizer for nonlinear unconstrained optimization problems. Wichrowska et al. (2017) extended this work by introducing a hierarchical RNN architecture which improves the generalization ability of the optimizer. Schramowski et al. (2018) demonstrated the benefit of a learned projection-free convex optimization algorithm, which relies on conditional gradients. Finally, Chen et al. (2017) cast the design of gradient-free black-bock optimization as a learning problem.

Furthermore, other approaches to solve differential equations using neural networks have been proposed, too. Lagaris et al. (1998) suggested to use a feed-forward neural network to approximate the solution of an ordinary or partial differential equation with initial or boundary conditions. Inspired by the Galerkin method, Sirignano & Spiliopoulos (2018) used a deep neural network to directly approximate the solution of a high-dimensional partial differential equation. E & Yu (2018) proposed the Deep Ritz method as a means to solve variational problems that arise from partial differential equations. Han et al. (2018) cast the problem of solving semilinear parabolic partial differential equations as a learning problem by using the reformulation of these differential equations as backward stochastic differential equations. In all of the above approaches, a neural network is used to help approximate the solution of the differential equation. The learning process corresponds to the numerical computation of a solution of the differential equation. In contrast, MLRK learns to improve the numerical integration process itself.

We proceed as follows. As the first step, *i.e.* identifying a good performance measure, we consider the general objective of step size control as well as the simplified and more practical objective that many controllers are based on. Then, we identify useful inputs by analyzing existing step size control algorithms. Furthermore, we show how local information about the ODE can be used as additional favorable inputs. Before concluding, we demonstrate empirically how our proposed controller can be used to improve step size control and investigate the benefit of different inputs.

## 2 INITIAL VALUE PROBLEMS AND RUNGE-KUTTA METHODS

We give a brief introduction to initial value problems and Runge-Kutta methods, a numerical method for solving these problems. Furthermore, we will recap standard step size controllers and point out their underlying assumptions.

**Basics of Initial Value Problems and Runge-Kutta Methods.** An ordinary differential equation describes a system that depends on one variable, often referred to as *time*. An *initial value problem* additionally provides initial values: $y' = g(t, y)\,, \quad y(t_0) = y_0$ , with a function $g : [a, b] \times \mathbb{R}^m \to \mathbb{R}^m$, $m \in \mathbb{N}$. As closed-form solutions can be very hard to find, numerical methods such as Runge-Kutta methods have been developed. Their mathematical underpinning is quite evolved (Butcher, 2008; Hairer et al., 2000; Hairer & Wanner, 2002). A brief recap can be found in the appendix.

For iterative solvers, the local truncation error—the error committed in a single step using step size $h$—constitutes an important tool to control the step size as it allows to access the fitness of a step size.

**Definition 2.1.** The *local truncation error* in step $n + 1$ is defined as

$$\text{err}_{n+1}(h) = \|y(t_n + h) - y_{n+1}\| = C(t_n)h^{p+1} + \mathcal{O}(h^{p+2}), \tag{1}$$

where $y_{n+1} = \Phi(t_n, y_n, h)$ is a single Runge-Kutta step of a $p$-th order method using step size $h$. We call $C(t_n)$ the *principal error term*.

We will also denote the local truncation error as *local error* and refer to the appendix for simple ways to estimate it. We denote the error estimates as $\text{err}_n$.

**The Objective of Step Size Control.** As numerical integration is used to approximate a solution, high accuracy is a desired property. However, in practice, computational resources might be limited and efficient use of these resources can be crucial in some applications. Hence, an efficient step size controller should maximize the accuracy of the approximate solution while minimizing the computational cost. These are competing goals and require a trade-off, which is achieved by considering the Lagrangian of the error $E(H)$ and the work cost $W(H)$ produced by a sequence of step sizes $H = (h_n)_{n=0}^{N-1}$, $E(H) + \lambda \cdot W(H)$. Butcher (2008) uses two integrals describing the error $E(H)$ and the work cost $W(H)$. Under some assumptions one can show that optimal step sizes w.r.t. this objective causes local errors equal to the Lagrange multiplier $\lambda$, which is often referred to as the tolerance parameter. Namely, we need to assume that the global error is equal to the sum of local errors and all step sizes are small enough so that the loss can be approximated by the integrals suggested by Butcher. However, these are quite strong assumptions. The optimal step size depends on the tolerance parameter, and hence, might be too large. Furthermore, the assumption that the global error is equal to the sum of the local errors is not true in general.

**Classical Step Size Control.** Most common step size control mechanisms aim to keep the local errors equal to a tolerance parameter so that a standard controller is based on the following idea: For a Runge-Kutta method of order $p$, the local error for step size $h$ at time $t$ is approximately $\text{err} \approx C(t)h^{p+1}$, and, accordingly, the desired optimal step size $h_{\text{opt}}$ is given by $h_{\text{opt}} \approx h \, (\text{tol}/\text{err})^{\frac{1}{p+1}}$. This formula is then used to adapt the step size as

$$h_{n+1} = r_n h_n, \qquad r_n = \max\left(\alpha, \min\left(\beta, \gamma\left(\frac{\text{tol}}{\text{err}_{n+1}}\right)^{\frac{1}{p+1}}\right)\right), \tag{2}$$

where a safety factor $0 < \gamma < 1$ and a minimal and maximal factor $\alpha, \beta \in \mathbb{R}_{\geq 0}$, $\alpha \leq \beta$ are included to avoid excessive step rejection and ensure a smooth step size control. Söderlind (2002) pointed out that the underlying assumptions of Eq. (2) are rather strong. It assumes slow variations in $C$ and requires $h$ to be sufficiently small so that it exhibits its theoretical asymptotic behavior. Both are not true in general.

**Control Theory on Step Size Control in Runge-Kutta Methods.** Gustafsson et al. (1988) discussed step size control in the context of a proportional-integral-derivative (PID) controller. The classical step size control mechanism in Eq. (2) can be regarded as an I-controller. They then demonstrated the oscillatory behavior of this controller when applied to certain problems. They report the poor stabilizing capability of an I-controller as the origin of these oscillations, which are further accentuated by a large integration gain. To overcome this, they Gustafsson et al. (1988) suggested a PI-controller, which in addition to the integral term used in a standard step size control, also includes a proportional term and can be expressed as

$$h_{n+1} = r_n h_n, \qquad r_n = \max\left(\alpha, \min\left(\beta, \gamma\left(\frac{\text{tol}}{\text{err}_{n+1}}\right)^{n_1}\left(\frac{\text{tol}}{\text{err}_n}\right)^{n_2}\right)\right). \tag{3}$$

**A Predictive Controller.** Both the classical step size controller, see Eq. (2), and the PI-controller in Eq. (3) rely on the assumption that the local truncation error can be described as a function of $h$ that remains independent of $t$. This is certainly not true for all cases. Therefore, Gustafsson (1994), e.g., discussed a controller based on a prediction of the principal error term. A simple model that assumes a constant linear trend in $\log C(t)$ was suggested. Implementing the predicted principal error leads to

$$h_{n+1} = \frac{h_n}{h_{n-1}}\left(\frac{\text{tol}}{\text{err}_{n+1}}\right)^{n_1}\left(\frac{\text{err}_n}{\text{err}_{n+1}}\right)^{n_2} h_n. \tag{4}$$

## 3 META-LEARNING RUNGE-KUTTA: NEURAL STEP-SIZE CONTROLLER

Step size controllers for Runge-Kutta are typically still designed by hand. Formulas such as Eq. (2), (3) and (4) require parameter fine-tuning. In this work, we take a different tack and instead suggest to replace these hand-designed update rules with a learned controller, hereinafter referred to as *optimizer*. The key steps to cast the design of a controller as a learning problem is to determine a good performance measure and appropriate inputs.

**Performance measure.** As the first step we need to identify an appropriate performance measure. As discussed, a trade-off between numerical accuracy and computational cost of the solution must be made, *e.g.* by considering the Lagrangian of error and work cost. The work cost is generally correlated with the number of integration steps. It is crucial to take the length of the integration interval $t_n - t_0$ into account as well, which leads to the following performance measure:

$$l^{\text{Lagrange}}(t_n, y_n, \text{err}_n) = \frac{\|y(t_n) - y_n\| + \text{tol} \cdot n}{t_n - t_0} \, . \tag{5}$$

In contrast, if we follow the concept of most step size controllers, an optimal step size is achieved if the local error is equal to the tolerance, which suggests the performance measure

$$l^{\text{local}}(t_n, y_n, \text{err}_n) = \|\text{err}_n - \text{tol}\| \, . \tag{6}$$

Both types of loss functions offer various strengths and weaknesses over one another.

The loss Eq. (5) incorporates the essential quantities that an efficient algorithm should minimize. However, the true global error of numerical integration is generally unknown and can only be approximated. In general this can be computationally demanding and as a result make the training process very expensive. Furthermore, when the global error is employed, optimal step sizes may depend on the integration interval as well, *e.g.* the choice of step size in the first few steps may have different effects on the global error at different points in the integration interval.

Classic step size controllers are typically based on local errors alone without regards to global strategies. The objective of these controllers is reflected by the loss Eq. (6). Although this objective minimizes the Lagrangian of the global error and the number of steps only under strong assumptions it does qualify as an appropriate practical objective for step size controllers. The advantage here is that estimates of the local error associated with a step size are available and no additional computation to determine the loss of a step size is required. This allows efficient training even when the analytical solution of an initial value problem is unknown.

**Input Features.** Existing step size controllers rely on error estimates to update the step size. Specifically, $\frac{\text{tol}}{\text{err}_n}$ is important, as evident in Eq. (2), (3) and (4). We intuitively expect a data driven approach to be able to utilize these features. This gives rise to the first set of input features,

$$\psi(\text{err}_{n+1}, \cdot) = \log\left(\frac{\text{tol}}{\text{err}_{n+1}}\right) \, . \tag{7}$$

Here, $\psi(\text{err}_{n+1}, \cdot)$ denotes the input for the optimizer with optional variables. Next, we want to investigate additional input features that may be beneficial. The principal error term as well as higher order terms of the local errors can be expressed with elementary differentials of the ODE, *c.f.* (Hairer et al., 2000, Section II.3). These elementary differentials are formed of function values and partial derivatives of $g$ and allow a Taylor approximation of the local error function $\text{err}_{n+1}(h) = \phi(t_n, h)h^{k+1} + \mathcal{O}(h^{k+2})$, where $\phi(t_n, h)$ is a polynomial in $h$ with coefficients formed from partial derivatives of $g$ up to order $k > p$. The roots of the polynomial $\phi(t_n, h)h^{k+1} - \text{tol}$ approximate the optimal step size. Moreover, it is reasonable to assume that the local error takes values greater tol for some $h > 0$, implying the existence of real roots. Since the complex roots of polynomials are known to depend continuously on the coefficients of the polynomial, *c.f.* (Rahman & Schmeisser, 2002, Theorem 1.3.1), it is clear that the optimal step size, a real root of $p(h)$, depends continuously on the partial derivatives and function values of $g$ as well as the tolerance tol. From the Universal Approximation Theorem (Hornik, 1991), it then follows that the optimal step size function can be approximated by a neural network with input $\left((\partial^\alpha g(t_n, y_n))_{|\alpha|=k}, \ldots, \partial g(t_n, y_n), g(t_n, y_n), \text{tol}\right)$. This suggests that the partial derivatives of $g$ can also be an appropriate input for our controller,

$$\psi(\text{err}_{n+1}, \cdot) = \left(\log\left(\frac{\text{tol}}{\text{err}_{n+1}}\right), (\partial^\alpha g(t_n, y_n))_{|\alpha|=p}, \ldots, \partial g(t_n, y_n), g(t_n, y_n)\right) \, . \tag{8}$$

However, providing partial derivatives requires additional computation whereas the $p$ function values that were computed during the Runge-Kutta step are conveniently available. As a trade-off between "perfect" information in the form of higher order partial derivatives and the computational effort to compute them, we also propose to use the following input:

$$\psi\left(\mathrm{err}_{n+1}, \cdot\right) = \left(\log\left(\frac{\mathrm{tol}}{\mathrm{err}_{n+1}}\right), g_n^{(1)}, \ldots, g_n^{(p)}\right), \tag{9}$$

where $g_n^{(1)}, \ldots, g_n^{(p)}$ denotes $p$ function values of the initial value problem.

In contrast to the traditional input (7), our novel inputs (8) and (9) both provide local information about the structure of the problem at hand, which allows one to keep the local error close to the tolerance parameter. Moreover, hand-designed controller and existing Runge-Kutta methods do not make use of these information, and it is difficult—if not impossible—to do so. Training an optimizer with input (8) or (9), as shown next, does so automatically and designs novel Runge-Kutta methods.

**Meta-Learner.** With a good performance measure and appropriate inputs at hand, we can now solve the step-size control problem as a meta-learning problem as sketched already in Fig. 1. Let the input for the optimizer be $x_n = \psi\left(\mathrm{err}_n, \cdot\right)$. We parameterize the optimizer $c$ using $\phi$ and update the step size as follows, $\log h_{n+1} = \log h_n + c_n\left(\psi(\mathrm{err}_{n+1}, \cdot), \phi\right)$. An approach in log space has the advantage that that we do not need to constrain the output of $c$ to be positive. Since $c_n$ corresponds to $\log r_n$, we will adjust the notation in a similar fashion. Due to their natural ability to handle sequential tasks, an LSTM was chosen in our experiments as optimizer $c$, and its prediction determines the step size which will be used in the next Runge-Kutta iteration. We denote the model of the optimizer, represented by the blue block in Fig. 1, by $m$, its parameters by $\phi$ and the hidden state by $\hat{h}_n$. It can be expressed by

$$\begin{pmatrix} \log r_{n-1} \\ \hat{h}_{n+1} \end{pmatrix} = m\left(x_n, \hat{h}_n, \phi\right). \tag{10}$$

The subsequent Runge-Kutta update in the yellow block of Fig. 1 first updates the step size

$$\log h_n = \log h_{n-1} + \log r_{n-1}, \tag{11}$$

and then uses it to execute the next Runge-Kutta step,

$$t_{n+1} = t_n + h_n, \qquad \begin{pmatrix} y_{n+1} \\ \mathrm{err}_{n+1} \end{pmatrix} = \Phi(t_n, y_n, h_n). \tag{12}$$

Here, $\Phi$ denotes the Runge-Kutta step from $y_n$ to $y_{n+1}$ as well as the error estimate $\mathrm{err}_{n+1}$ that is computed during that step. Afterwards, the next input for the model will be computed.

**Learning Objective.** Different initial value problems can require very different step sizes. Using an optimizer that is specialized for a certain class of problems allows to exploit the structure of these problems. The behavior of an initial value problem is described by a function $g$, therefore we can represent a class of initial value problems with a distribution over the functions $g$. Thus, an optimizer can be considered optimal for a class of problems, if it minimizes the expected loss:

$$L(\phi) = \mathbb{E}_g\left(\sum_{n=1}^N l(t_n, y_n, \mathrm{err}_n)\right), \tag{13}$$

where $t_n, y_n, \mathrm{err}_n$ were computed according to Eqs. (10), (11), (12). Here, $l$ denotes the loss function defined for an approximation $y_n$ at time point $t_n$ and the local error estimate $\mathrm{err}_n$. Note that our training loss directly corresponds to the performance measure we are interested in. The model $m$ can then learn the behavior of the given class of problems and use this knowledge to generalize to new examples of the same class and new, unseen classes. Specifically, since the provided performance measure $l$ is differentiable a.e., we can optimize the learning objective Eq. (13) using gradient descent on the parameters $\phi$. An estimate of the gradient $\frac{\partial L(\phi)}{\partial \phi}$ can be computed by sampling a function $g$ from the distribution of the class of initial value problems and applying backpropagation, *c.f.* (Rumelhart et al., 1986).

## 4 EXPERIMENTAL EVIDENCE

Our intention here is to investigate the benefits of Meta-Learning Runge-Kutta (MLRK). To this end we conducted experiments with different classes of initial value problems. Specifically, we

Table 1: On the test set of class (low-freq), MLRK was considerably faster (less many steps) than the baseline controller on average, while causing only a mildly larger mean global error at the end of the integration interval. Both the baseline controller and MLRK showed a gradually increasing mean global error at the end of the integration interval over the test set consisting of 1500 harmonic oscillators of class (med-freq).

| interval | (low-freq) | | | | (med-freq) | | | |
|---|---|---|---|---|---|---|---|---|
| | steps | | error | | steps | | error | |
| | Baseline | MLRK | Baseline | MLRK | Baseline | MLRK | Baseline | MLRK |
| 1 | 3.42 | 3.15 | 0.000004 | 0.000009 | 26.24 | 7.95 | 0.001588 | 0.008253 |
| 3 | 7.59 | 6.05 | 0.000017 | 0.000119 | 87.16 | 29.63 | 0.001686 | 0.011236 |
| 5 | 11.76 | 8.22 | 0.000032 | 0.000366 | 148.05 | 53.44 | 0.001739 | 0.012904 |
| 7 | 15.80 | 10.23 | 0.000048 | 0.000668 | 208.95 | 77.54 | 0.001735 | 0.014222 |
| 10 | 21.92 | 13.15 | 0.000073 | 0.001171 | 300.32 | 113.82 | 0.001816 | 0.016546 |

| interval | (high-freq) | | | |
|---|---|---|---|---|
| | steps | | error | |
| | Baseline | MLRK | Baseline | MLRK |
| 1 | 47.15 | 12.08 | 0.026415 | 0.085082 |
| 3 | 157.58 | 53.42 | 0.023223 | 0.081219 |
| 5 | 268.03 | 96.48 | 0.025230 | 0.091109 |
| 7 | 378.42 | 139.69 | 0.026177 | 0.094129 |
| 10 | 544.05 | 204.57 | 0.024858 | 0.094562 |

Table 2: The mean global errors of the baseline stay approximately the same. The optimizer trained on problem instances of class (low-freq) shows only a very gradual increment in the global error on instances of class (high-freq).

investigated our suggested loss functions and our theory that suggests that providing higher order partial derivatives of $g$ can be used as a means to anticipate the evolution of the local error and, hence, may lead to an improved performance of the optimizer. For a class of initial value problems we assume a parametric form of $g$ with a distribution over the parameters; details can be found in the appendix. Our datasets are obtained by sampling from these distributions. Finally, we compare our method to the baseline controller given by Eq. (2).

**Harmonic Oscillator.** First, we started with a simple class of ODEs. We trained MLRK on low varied harmonic oscillators. The corresponding results for training on high frequencies are similar and can be found in the appendix. As inputs we only used the error estimates, Eq. (7). We started with the dataset of harmonic oscillators with low frequencies (low-freq): the *training set* contained 30,000 instances, the *validation* set and *test set* each 1,500 instances. To measure the performance we considered the $L_1$-Lagrangian loss Eq. (5) as well as the number of steps needed for an integration interval and the corresponding error separately. Since the Lagrangian represents a trade-off between the two it is a particularly good measure.

Fig. 2(a) shows the mean loss during the integration of the test set, which consists of 1500 harmonic oscillators of class (low-freq). The solid lines show the mean loss in step $n$, the colored areas mark the standard deviation. MLRK achieves a smaller mean loss than the baseline, which confirms its ability to generalize to new problem instances of the same class of problem. When we compare the mean number of steps and the mean global error of MLRK and the baseline controller in Tab. 1, the necessity for a trade-off between the two objectives becomes clear. While the baseline needs on average more steps to complete the integration, it achieves a smaller error, the contrary is true for our optimizer. This makes it hard to compare the two methods based on just these values.

**Generalization Capability of Optimizer based on Low Frequency Harmonic Oscillators.** By evaluating the controller designed by MLRK on different test sets, we investigated its ability to transfer knowledge to problem instances of other classes. Fig. 2(b) shows that MLRK maintains a smaller mean loss on 1500 harmonic oscillators of class (med-freq). These oscillators are both higher in frequency and amplitude than the problem instances contained in the training set. This indicates that MLRK is able to generalize to these problems. Tab. 1 reveals that MLRK uses on average about a third of the integration steps the baseline needs for the tested integration intervals. The resulting mean global errors are indeed larger than those of the baseline controller, but still very reasonable. Moreover, evaluating MLRK on 1500 harmonic oscillators of class (high-freq) showed a mean loss similar to that of the baseline controller in Fig. 2(c). This suggests that MLRK generalizes to problem

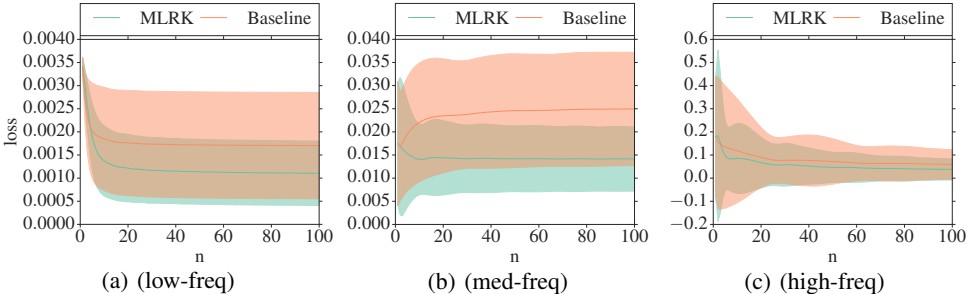

Figure 2: The mean $L_1$-Lagrangian loss (5) of the approximation $(t_n, y_n)$ in step $n$ over three different test sets. (a) MLRK achieved a smaller mean loss, thus, it generalizes well to new problem instances of the same class of problems (low-freq) it was trained on. (b) MLRK achieves a lower loss on instances with both higher amplitude and frequency than the problem instances used during training. Thus, MLRK is able to generalize to different problems. (c) Even on higher frequencies than the ones in (med-freq), MLRK shows a mean loss similar to that of the baseline controller.

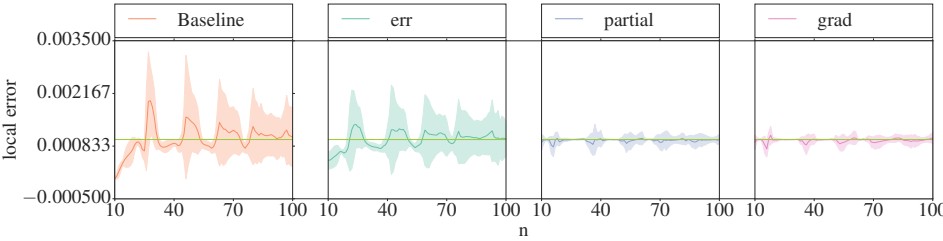

Figure 3: The mean local error over the test set of van der Pol ODEs for different methods is shown. The optimizer *err* was trained with input (7), optimizer *partial* was trained with input (8) and *grad* was trained with input (9). Providing local information such as those in input (8) and (9) results in increased responsiveness of the optimizer.

instances of this class as well. In Tab. 2 we observe a similar situation as in the previous experiment. The global error of the baseline controller stays approximately the same for different interval lengths, while the global error of MLRK increases very gradually, while using only a fraction of iterations.

**van der Pol.** "*I have a theory that whenever you want to get in trouble with a method, look for the van der Pol equation*" – P. Zadunaisky, 1982, *c.f*. (Hairer et al., 2000).

Next we conducted an experiment regarding the local $L_1$ loss (6). Experiments with harmonic oscillators revealed the optimizers ability to keep the local errors close to the tolerance parameter (*c.f*. appendix). Keeping the local error equal to a constant tolerance is not a hard problem for simple harmonic oscillators, therefore we turned towards *van der Pol* oscillators. We considered the following dataset of van der Pol oscillators of class vdP$(0, 1)$: The *training set* contained 50,000 instances, the *validation set* and the *test set* each 1,500 instances. The local errors of van der Pol oscillators vary drastically (*c.f*. Fig. 5, appendix). High spikes occur when the solution changes from being driven to being damped. As the behavior of van der Pol equations are more complex compared to harmonic oscillators, we expect that providing additional information about the problem such as partial derivatives or function values of $g$ can be beneficial. To investigate this, we considered MLRK with different inputs. One optimizer is only provided with the error estimate Eq. (7), another is provided with the error estimate and all partial derivatives Eq. (8) and the last is provided with the error estimate and the function evaluations that were computed during the Runge-Kutta step, Eq. (9).

The results are summarized in Fig. 3. Using *err* shows reoccurring high spikes similar to the baseline, although the spikes of the optimizer are lower on average. In contrast, MLRK with additional information on the partial derivatives (*partial*) shows a much smoother sequence of local errors. This demonstrates that the additional information in the partial derivatives of $g$ improves the ability of MLRK to anticipate the variations of the local errors and respond with adequate step sizes. MLRK with error estimates as well as function values of $g$ (*grad*) produces a smooth step size sequence

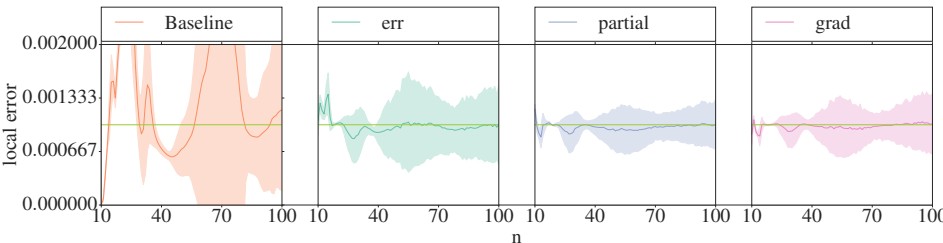

Figure 4: The mean local error over the test set of double pendulums for different methods is shown. The optimizer *err* was trained with input (7), optimizer *partial* was trained with input (8) and *grad* was trained with input (9). In all three cases MLRK is able to reduce the spikes in the local errors drastically, the use of local information reduces the variance of the local errors even further.

similar to the optimizer *partial*. This shows that the optimizer with additional input-information from $g$ are actually outperforming the baseline. The reoccurring spikes in the local error seem to decrease in amplitude with time. This is largely due to an averaging effect – for different values of $\sigma$ these spikes occur at different integration steps. All three methods reveal a reduction in *both* the mean number of steps and the mean local errors, $c.f.$ Tab. 3, appendix.

**Double Pendulum.** To demonstrate the benefit of MLRK in an application we conducted experiments with double pendulum ODEs ($c.f.$ appendix). The chaotic behaviour of the double pendulum leads to sudden spikes in the local error, a particularly challenging problem for step size controllers. To investigate the ability of MLRK to avoid these spikes by adjusting the step size appropriately, we consider the $L_1$ local loss as in the previous experiment and a dataset of double pendulums of class (pendulum). The *training set* contained 50,000 instances, the *validation set* and the *test set* each 1,500 instances. Again, the different inputs Eqs. (7), (8) and (9) were examined.
The results are shown in Fig. 4 and reveal the baseline controllers inability to produce steady local errors. Here, the benefit of a learned update rule over a static one becomes evident. The optimizer *err* that is only provided with the local error estimates is able to reduce the spikes in the local errors to a remarkable degree. Moreover, local information about the problem allows the optimizers *partial* and *grad* to reduce the variance in the local errors even further.

## 5 CONCLUSION

We have shown how to cast the design Runge-Kutta methods for solving initial value problems as a learning problem. We established appropriate performance measures and useful inputs for the controller, which constitute the key ingredients of the resulting Meta-learning Runge-Kutta (MLRK), which learns step size controllers that are specialized to a particular class of initial value problems. Our experimental results demonstrate that MLRK can indeed learn to design novel Runge-Kutta methods that perform better than a hand-designed Runge-Kutta approach. Furthermore, we observed a remarkable degree of generalization to other classes of initial value problems. More importantly, examining the effect of different inputs demonstrated that the additional information contained in the function values and partial derivatives of $g$ leads to a substantial improvement in performance of the automatically designed solvers.

There are several interesting avenues for future work. While MLRK generalizes well to problem instances of the same class and even to problems of similar classes, when confronted with a very different type of problem, MLRK does not generalize well yet. Wichrowska et al. (2017) showed how a carefully chosen network architecture and a diverse training set improves generalization of their optimizer to many different classes of optimization problems. A similar approach may lead to improved generalization of MLRK. Another common aspect of Runge-Kutta is the need for step rejection in case the local error exceeds the tolerance. The problem here is that a single large error can in general not be compensated for, even if subsequent step sizes are chosen very small. In this case, a step is usually rejected and repeated using a smaller step size. Here the challenge is to choose a step size small enough to meet the accuracy requirements but at the same time not too small, since the choice of step size influences the entire step size sequence to come.

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

## A APPENDIX

### RUNGE-KUTTA METHODS

The mathematical theory of Runge-Kutta methods is quite evolved. We will only recap a few basics here, for further details see for example (Butcher, 2008; Hairer et al., 2000; Hairer & Wanner, 2002).

**Definition A.1.** Let $s \in \mathbb{N}$, $a_{ij} \in \mathbb{R}$ for $i, j \in \{1, \dots, s\}$, $c_i \in \mathbb{R}$ for $i \in \{1, \dots, s\}$ and $b_i \in \mathbb{R}$ for $i \in \{1, \dots, s\}$. Let $g : \mathbb{R} \times \mathbb{R}^m \to \mathbb{R}^m$ be a function that describes an initial value problem. The method defined by

$$y_{n+1} = y_n + h \sum_{i=1}^{s} b_i k_i \,, \tag{14}$$

$$k_i = g(t_n + h c_i, y_n + h \sum_{j=1}^{s} a_{ij} k_j) \,, \tag{15}$$

for step size $h$ is called an *s-stage Runge-Kutta method*. If $a_{ij} = 0$ for $i \leq j$, the method is called *explicit*, otherwise it is called *implicit*.

**Definition A.2.** A Runge-Kutta method is of *order $p$* if for sufficiently smooth initial value problems

$$\|y(t_0 + h) - y_1\| \leq K h^{p+1}$$

holds.

Note that a higher order method yields more accurate results, consequently a high order is a desired characteristic of a Runge-Kutta method. However, higher orders can only be achieved by the use of more stages. For an $s$-stage Runge-Kutta method the order $p$ is bounded by the number of stages, $p \leq s$. Up to order 4 there exist methods with $p = s$, for order 5 and higher $p$ is strictly smaller than $s$, $c.f.$ (Butcher, 2008, Theorem 324B). The order of an $s$-stage Runge-Kutta method depends on the coefficients $a_{ij}$, $b_i$, $c_i$. Order conditions for these coefficients have been developed, which ensure a certain order of the method.

The problem of step size control can be described in the following way: Based on some input values, $e.g.$ $t_n, y_n, \mathrm{err}_n$ and possibly additional characteristics of $g$, we must choose a step size $h_n$, which is then used to evaluate $g$ at $p$ different points. These points are determined by $h_n$, see Definition A.1. The resulting stages in Eq. (15) are then used to compute the approximation $y_{n+1}$ of $y(t_n + h_n)$, $c.f.$ Eq. (14). The problem is to choose $h_n$ in a way such that the approximation $y_{n+1}$ fulfills some desired properties. For example we may require that the local error of $y_{n+1}$ is close to some tolerance parameter. In fact, this is the standard objective of most common step size controllers.

### ERROR ESTIMATION

The local errors are of particular importance for step size control and can fortunately be estimated very efficiently. To that end, we take a look at a simple idea as described in (Butcher, 2008, p. 198): Suppose we have two approximations for $y(t_n)$ of order $\hat{p}$ and $\tilde{p}$ respectively, that is

$$\hat{y}_n = y(t_n) + \mathcal{O}(h^{\hat{p}+1}),$$
$$\tilde{y}_n = y(t_n) + \mathcal{O}(h^{\tilde{p}+1}),$$

where $y(t)$ is the solution of

$$y'(t) = g(t, y(t)), \quad y(t_{n-1}) = y_{n-1},$$

and the step size $h$ that was used to obtain both approximations is given by

$$h = t_n - t_{n-1}.$$

If $\hat{p} > \tilde{p}$,

$$\hat{y}_n - \tilde{y}_n = y(t_n) - \tilde{y}_n + \mathcal{O}(h^{\tilde{p}+2})$$

can be used as an approximation for the local truncation error of the $\tilde{p}$-th order Runge-Kutta method:

$$\mathrm{err}_n(h) \approx \|\hat{y}_n - \tilde{y}_n\|.$$

To obtain error estimates each step has to be carried out with two methods of different orders. Here, the easiest approach is to use two separate Runge-Kutta methods to compute $\hat{y}$ and $\tilde{y}$, however this is quite costly, especially for implicit methods. A more efficient approach to obtain error estimates are the embedded Runge-Kutta methods. Here both approximation use the same stages $k_i$ in Eq. (15) but two different pairs of coefficients $b_i$, $\hat{b}_i$ in Eq. (14). By doing so, the stages can be reused and an error estimate can be obtained with a cheap extra linear combination of the $k_i$'s. We denote the error estimates by

$$\mathrm{err}_n = \|\hat{y}_n - \tilde{y}_n\|. \tag{16}$$

### CLASSES OF INITIAL VALUE PROBLEMS

For loss functions such as the one in Eq. (5) it is useful to consider initial value problems with known solutions so that we can compare the numerical approximation to the true function values of the solution. For this reason we considere simple harmonic oscillators and linear differential equations with constant coefficients. If we employ loss functions as Eq. (6), local rather than global errors are of interest. For these loss functions it is interesting to consider initial value problems for which the local errors vary drastically during the integration such as van der Pol oscillators.

SIMPLE HARMONIC OSCILLATOR

A simple harmonic oscillator is a harmonic oscillator that is neither driven nor damped and can be characterized by

$$m\,x''(t) = -k\,x(t),$$

where $m$ is the mass of the oscillator, $x$ the position and $k$ describes the restoring force that is applied, when displaced from its equilibrium position. It can be transformed to

$$y_1'(t) = y_2(t)\,,$$
$$y_2'(t) = -\frac{m}{k}y_1(t)\,.$$

The initial values provide the initial position $x(t_0)$ of $x$ and the initial velocity $x'(t_0)$,

$$y_1(t_0) = x(t_0),$$
$$y_2(t_0) = x'(t_0).$$

The solution of the simple harmonic oscillator takes the form

$$x(t) = A\,\cos(\omega t + \varphi), \quad \omega = \sqrt{\frac{m}{k}},$$

where $A$ and $\varphi$ are uniquely determined by the initial values. The frequency of the oscillations is determined by $\omega$, $A$ is the magnitude of the oscillations and $\varphi$ is a phase-shift.

A class of this kind of initial value problem can be described by a distribution over $\frac{m}{k}$ and the initial values or alternatively a distribution over $A$, $\omega$ and $\varphi$. The following classes of initial value problems were used in the experiments

$$
\begin{array}{llll}
A \sim U(0,5), & \omega \sim U(0,10), & \varphi \sim U(0,1), & \text{(high-freq)} \\
A \sim U(0,5), & \omega \sim U(0,20), & \varphi \sim U(0,1), & \text{(higher-freq)} \\
A \sim U(0,1), & \omega \sim U(0,1), & \varphi \sim U(0,1), & \text{(low-freq)} \\
A \sim U(1,5), & \omega \sim U(1,5), & \varphi \sim U(0,1), & \text{(med-freq)}
\end{array}
$$

where $U(a,b)$ denotes the uniform distribution over the interval $[a,b]$.

LINEAR CONSTANT COEFFICIENT

A linear differential equation with constant coefficients can be described by

$$y'(t) = A\,y(t)\,, \qquad y(t_0) = y_0\,,$$

where $A \in \mathbb{R}^{m \times m}$. The solution of this differential equation is known and can be expressed in terms of the eigenvalues $\lambda_1, \ldots, \lambda_m$ with corresponding eigenvectors $v_1, \ldots, v_m$ of the matrix $A$:

$$y(t) = \sum_{j=1}^{m} c_j\,e^{\lambda_j t}\,v_j,$$

with coefficients $c_j \in \mathbb{R}$ that are uniquely determined by the initial values $y(t_0) = y_0$. The initial value problem is stable, if $\mathrm{Re}(\lambda_j) < 0$ for all $j \in \{1, \ldots, m\}$ and unstable if $\mathrm{Re}(\lambda_j) > 0$ for some $j \in \{1, \ldots, m\}$. The oscillatory behavior of the problem is determined by $\mathrm{Im}(\lambda_j)$. Note that the simple harmonic oscillators are linear differential equations with constant coefficients, where $A \in \mathbb{R}^{2 \times 2}$, $\mathrm{Re}(\lambda_j) = 0$ for $j \in \{1, 2\}$ and $\mathrm{Im}(\lambda_1) = -\mathrm{Im}(\lambda_2) = \sqrt{\frac{m}{k}}\,i$.

For the experiments we used differential equations with $A \in \mathbb{R}^{3 \times 3}$. The matrix $A$ either has three real eigenvalues $\lambda_1, \lambda_2, \lambda_3 \in \mathbb{R}$ or one real eigenvalue $\lambda_1 \in \mathbb{R}$ and two complex eigenvalues $\lambda_2 = a + bi$, $\lambda_3 = a - bi$, with $a, b \in \mathbb{R}$. In this case we choose $c_2 = c_3$ to obtain a real solution. We can specify a class of linear constant coefficient differential equations by a distribution on the eigenvalues of $A$ and the coefficients $c_j$. The classes that were used for the experiments were described by the distributions

$$
\begin{array}{llll}
\lambda_1 \sim U(-2.5,-0.1), & a \sim U(-2.5,-0.1), & b \sim U(0,5), & \text{(stable)} \\
\lambda_1 \sim U(0,1), & a \sim U(-2.5,-0.1), & b \sim U(0,5), & \text{(unstable)} \\
\lambda_1 \sim U(-1,0), & a \sim U(0,1), & b \sim U(1,2), & \text{(osc-increasing)}
\end{array}
$$

where $U(a,b)$ denotes the uniform distribution on the interval $[a,b]$.

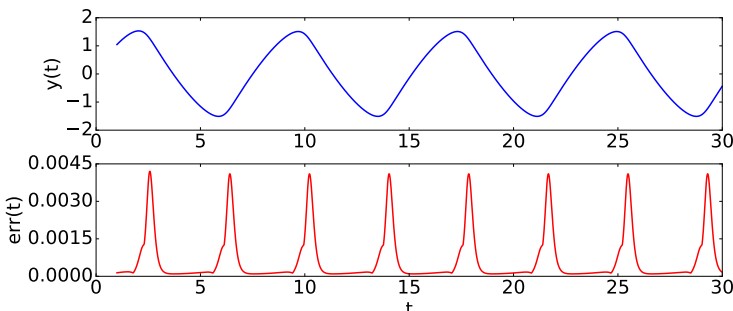

Figure 5: The upper plot shows the numerically approximated solution $y_1(t)$ of a van der Pol oscillator with $\sigma = 2$. The lower plot displays the local error estimates for $h = 0.1$. The error estimates show high spikes whenever the solution switches from being driven to being damped.

### VAN DER POL OSCILLATOR

*Van der Pol oscillators* arise in the study of limit cycles. A *van der Pol oscillator* can be described by

$$y_1'(t) = y_2, \qquad\qquad\qquad y_1(0) = 2,$$
$$y_2'(t) = \sigma(1 - y_1^2)y_2 - y_1, \qquad\qquad\qquad y_2(0) = 0,$$

with $\sigma > 0$. For $\sigma = 2$ an approximate solution is displayed in Figure 5. Small oscillations are amplifies and large oscillations damped, *c.f.* (Hairer et al., 2000, pp. 111). The local errors of a van der Pol oscillator show high variations. For the experiments we used van der Pol oscillators with the following distributions

$$\sigma \sim U(0, 1), \qquad\qquad\qquad\qquad (\text{vdP}(0,1))$$
$$\sigma \sim U(1, 2). \qquad\qquad\qquad\qquad (\text{vdP}(1,2))$$

### DOUBLE PENDULUM

A *double pendulum* is a pendulum attached to another pendulum, *c.f.* Fig. 6. The length of the first and second pendulum are given by $L_1$ and $L_2$, respectively, their masses by $M_1$ and $M_2$ and the angle by $\theta_1$ and $\theta_2$. The equations of motion for the double pendulum can be written in different forms, we chose the following formulation in terms of the angular acceleration:

$$\ddot{\theta}_1 = \frac{M_2 L_1 \dot{\theta}_1^2 \sin(\theta_2 - \theta_1) \cos(\theta_2 - \theta_1) + M_2 g \sin(\theta_2) \cos(\theta_2 - \theta_1)}{L_1(M_1 + M_2 - M_2 \cos^2(\theta_2 - \theta_1))}$$
$$+ \frac{M_2 L_2 \dot{\theta}_2^2 \sin(\theta_2 - \theta_1) - (M_1 + M_2) g \sin(\theta_1)}{L_1(M_1 + M_2 - M_2 \cos^2(\theta_2 - \theta_1))}$$
$$\ddot{\theta}_2 = \frac{(M_1 + M_2)\left(g \sin(\theta_1) \cos(\theta_2 - \theta_1) - L_1 \dot{\theta}_1^2 \sin(\theta_2 - \theta_1) - g \sin(\theta_2)\right)}{L_2(M_1 + M_2 - M_2 \cos^2(\theta_2 - \theta_1))}$$
$$- \frac{M2 L_2 \dot{\theta}_2^2 \sin(\theta_2 - \theta_1) \cos(\theta_2 - \theta_1)}{L_2(M_1 + M_2 - M_2 \cos^2(\theta_2 - \theta_1))}$$

The initial angular position and velocity $\theta_1, \theta_2, \dot{\theta}_1, \dot{\theta}_2$ determine the trajectory of the double pendulum. For the experiments we used double pendulums with the following distribution

$$M_1 \sim U(0.5, 1), M_2 \sim U(0.5, 2), L_1 \sim U(0.5, 1), L_2 \sim U(0.5, 1). \qquad (\text{pendulum})$$

### ADDITIONAL EXPERIMENTS

### ON VAN DER POL

Table 3 shows the mean number of steps, local error and wall clock time over 1500 van der Pol equations of class (vdP$(0, 1)$).

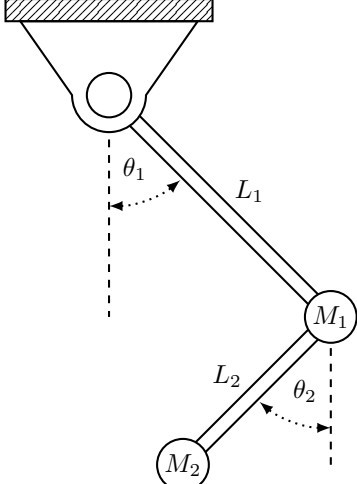

Figure 6: A double pendulum is one pendulum attached to another pendulum. The masses $M_1$ and $M_2$ of these pendulums may vary as do the lengths $L_1$ and $L_2$. The initial configuration of angular position and velocity determines the trajectory of the double pendulum.

HIGH FREQUENCY HARMONIC OSCILLATORS

In this experiment, we trained a controller specialized for simple harmonic oscillators of class (high-freq) which were described in Sec. A. Thus we chose the following datasets:

| | |
|---:|:---|
| **Training set**: | 30000 Harmonic oscillators of class (high-freq). |
| **Validation set**: | 1000 Harmonic oscillators of class (high-freq). |
| **Test set**: | 1000 Harmonic oscillators of class (high-freq). |

As loss we chose the $L_1$-Lagrangian Eq. (5). As inputs we used only the error estimates, Eq. (7). The loss of the learned controller and the baseline controller, averaged over the test dataset, is shown in Fig. 7(a). Both controllers are used to execute 100 integration steps for each instance of the test dataset. Subsequently, the loss Eq. (5) is computed for each approximation $(t_n, y_n)$. The solid lines show the mean loss in step $n$, the colored areas mark the standard deviation. Here, the loss of our learned controller is on average smaller than the loss of the baseline controller, which demonstrates the controllers generalization capability to new problems of the same class.

Tab. 4 summaizres the global error at the end of the integration interval and the number of steps needed during the integration for different interval lengths in range 1 to 10. All values are averaged over the test dataset. Here, the baseline controller needs much more steps than the learned controller, but maintains a smaller error. In fact, the error caused by the baseline is approximately the same for all shown interval lengths, while the error caused by the trained model increases with interval length. This is due to the assumption that the global error is the sum of the local errors. In particular, the loss Eq. (5) rates step size sequences with a higher global error better, if they can balance the higher error by achieving a longer integration interval.

While the general assumption that the global error increases with the length of the integration interval is often fulfilled, the specific assumption of a linear accumulation of the global error, as in our loss, is usually not fulfilled. This is a good example for why the assumptions leading to the widely used objective to keep the local errors constant are too strong.

Table 3: The mean number of steps, local error and wall clock time over 1500 van der Pol equations are shown for different lengths of the integration interval. Our method *err* is slightly faster and uses less steps than the baseline while producing smaller local errors during the integration, see also Figure 3. While *partial* and *grad* reduce the number of steps even further, wall time is increased. However one can clearly see that *partial* and *grad* outperform both the baseline and *err* regarding the local error.

| int | steps | | | |
|---|---|---|---|---|
| | baseline | err | partial | grad |
| 1 | 21.59 | 16.42 | 12.40 | 12.09 |
| 3 | 33.74 | 29.12 | 25.16 | 24.74 |
| 5 | 45.43 | 41.07 | 36.42 | 36.02 |
| 7 | 56.84 | 52.57 | 48.34 | 47.97 |
| 10 | 73.46 | 69.41 | 65.40 | 65.04 |

| int | mean local error | | | |
|---|---|---|---|---|
| | baseline | err | partial | grad |
| 1 | $7.17e-4$ | $6.58e-4$ | $4.01e-4$ | $3.74e-4$ |
| 3 | $6.40e-4$ | $5.45e-4$ | $2.49e-4$ | $2.28e-4$ |
| 5 | $5.18e-4$ | $4.47e-4$ | $1.95e-4$ | $1.75e-4$ |
| 7 | $4.97e-4$ | $4.16e-4$ | $1.57e-4$ | $1.39e-4$ |
| 10 | $4.59e-4$ | $3.82e-4$ | $1.32e-4$ | $1.18e-4$ |

| int | time | | | |
|---|---|---|---|---|
| | baseline | err | partial | grad |
| 1 | 0.0255 | 0.0263 | 0.0254 | 0.0221 |
| 3 | 0.0405 | 0.0375 | 0.0460 | 0.0403 |
| 5 | 0.0591 | 0.0517 | 0.0681 | 0.0596 |
| 7 | 0.0858 | 0.0742 | 0.1036 | 0.0897 |
| 10 | 0.0971 | 0.0825 | 0.1201 | 0.1065 |

| interval | steps | | error | |
|---|---|---|---|---|
| | Baseline | Optimizer | Baseline | Optimizer |
| 1 | 45.50 | 13.73 | 0.023092 | 0.030499 |
| 3 | 152.11 | 43.12 | 0.022266 | 0.040429 |
| 5 | 258.73 | 74.40 | 0.022731 | 0.049529 |
| 7 | 365.30 | 106.31 | 0.022011 | 0.055821 |
| 10 | 525.18 | 154.25 | 0.023353 | 0.070451 |

Table 4: The mean values over the test set consisting of 1000 harmonic oscillators of class (high-freq) are shown. While the baseline needs much more steps than our trained model, it achieves a smaller global error. The global error of the baseline stays approximately the same for all integration intervals.

It is also interesting to evaluate the learned controller on other classes of initial value problems. Fig. 7(b) shows the mean loss of the learned controller and the baseline controller over a test set consisting of 1500 harmonic oscillators of class (higher-freq). This class contains oscillators of class (high-freq) but also includes higher frequency ones. The learned controller has a lower mean loss than the baseline controller which indicates that it is able to transfer knowledge from problems of class (high-freq) to problems of class (higher-freq).

When we compare the number of steps and the global error at the end of the integration interval, see Tab. 5, we can observe a similar situation as in Tab. 4. The learned controller needs less steps than the baseline controller. The global error however stays approximately the same for the baseline controller while it increases with interval length for the learned controller.

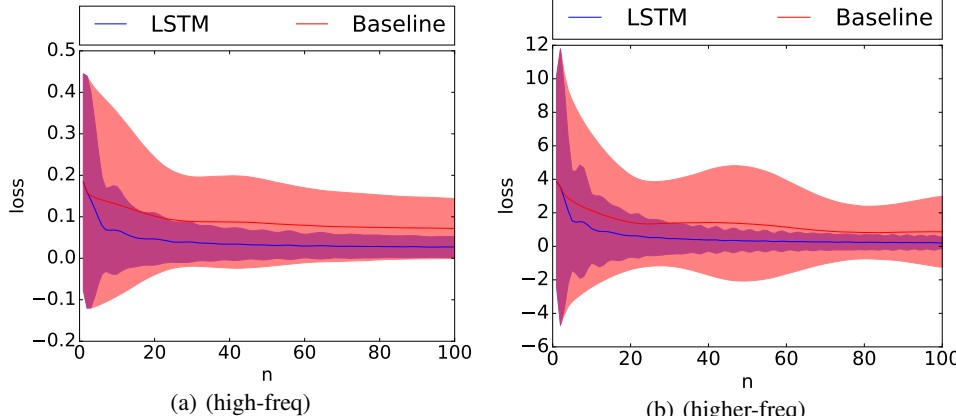

(a) (high-freq)  (b) (higher-freq)

Figure 7: The red and blue line show the mean loss Eq. (5) of the approximation $(t_n, y_n)$ in step $n$ over the test set. The red and blue area indicate the standard deviation. (a) The learned model LSTM attains a smaller mean loss on new instances of the same class of problems (high-freq) it was trained on, this is especially pronounced in the steps $n \geq 5$. (b) The test set consists of 1500 harmonic oscillators of class (higher-freq), which means that it also contains higher frequency oscillators. The mean loss of our learned controller is lower than the mean loss of the baseline controller which indicates that our method generalizes to these different problem instances.

| interval | steps | | error | |
|---|---|---|---|---|
| | Baseline | Optimizer | Baseline | Optimizer |
| 1 | 116.76 | 22.26 | 0.453698 | 0.511174 |
| 3 | 386.40 | 75.64 | 0.442521 | 0.620779 |
| 5 | 656.06 | 141.29 | 0.426773 | 0.770457 |
| 7 | 925.80 | 212.19 | 0.418467 | 0.925463 |
| 10 | 1330.31 | 319.85 | 0.413902 | 1.427778 |

Table 5: The mean number of steps and the mean global error at the end of the integration interval was computed over a test set of 1500 harmonic oscillators of class (higher-freq). While the baseline controller needs much more steps than our learned controller, it has on average a lower global error, which stays approximately the same for different interval lengths.

HARMONIC OSCILLATOR

We learn a controller optimized for simple harmonic oscillators. We choose the following data sets:

**Training set**: 30000 Harmonic oscillators of class (low-freq).
**Validation set**: 1500 Harmonic oscillators of class (low-freq).
**Test set**: 1500 Harmonic oscillators of class (low-freq).

First, we evaluate our method on a test set consisting of the same class of initial value problems the controller was trained on. Figure 8(a) shows the mean local error of the learned controller and the baseline on this test set. Both controllers are capable to keep the local errors close to the tolerance. Our method is slightly closer to the tolerance. Next, we evaluate our learned controller on different test sets. When we test the learned controller on harmonic oscillators with higher frequencies and higher amplitude, we find that our method generalizes to these problem instances as well. In fact, both the learned and the baseline controller are able to keep the local errors close to the tolerance as displayed in Figure 8(b). Similar results were found for harmonic oscillators of the class (med-freq). Hence, our learned controller is able to transfer knowledge from oscillators with low frequencies to oscillators with higher frequencies.

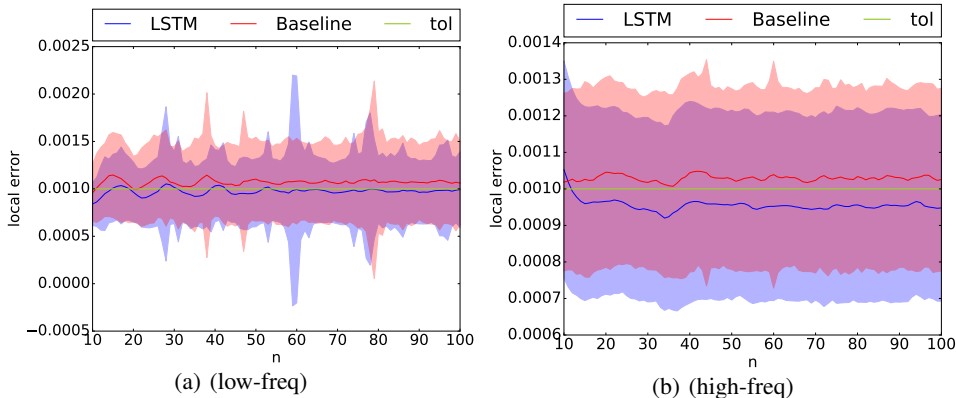

(a) (low-freq)  (b) (high-freq)

Figure 8: The mean local error of our learned controller and the baseline controller is shown. (a) Testing on 1500 harmonic oscillators of class (low-freq) shows that both the baseline and the learned controller are able to keep the local errors close to the tolerance. The mean local error of the learned controller is slightly closer to the tolerance. (b) For a test set consisting of 1500 harmonic oscillators of class (high-freq) the two controllers show a similar performance, both are able to keep the local error close to the tolerance. Testing the controllers on problem instances of class (med-freq) yields a similar results.

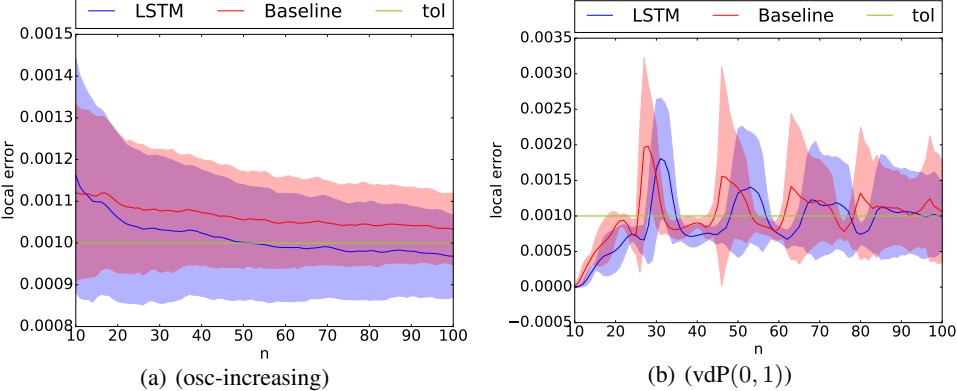

(a) (osc-increasing)  (b) (vdP$(0, 1)$)

Figure 9: The mean local error over different test sets is shown. (a) The test set consists of 1500 linear differential equations with constant coefficients of class (osc-increasing). For these problem instances the learned controller demonstrates its ability to keep the mean local error close to the tolerance. In fact, the local errors of our controller appear to be closer to the tolerance than the local errors of the baseline for some part of the integration. (b) Testing our controller on a set of 1500 van der Pol oscillators of type (vdP$(0, 1)$) reveals a similar performance to that of the baseline controller. Both controllers show high spikes in the local errors. These occur whenever the solution changes from being driven to being damped, see Figure 5.

We also want to evaluate the generalization capability of our method to problem instances of different classes of initial value problems. Our method shows an acceptable performance on linear differential equations with constant coefficients of the class (osc-increasing), see Figure 9(a). Here, the performance of our controller is similar to the baseline controller. The learned controller obtains local errors which are closer to the tolerance in some steps. Figure 9(b) shows a similar performance of our controller to the baseline controller on van der Pol oscillators. Both methods show high spikes in the local errors. These occur whenever the solution of a van der Pol oscillator changes from being driven to being damped as demonstrated in Figure 5. Our controller does not react quickly to the sudden changes in the local error of van der Pol equations. This behavior is very different from harmonic oscillators, thus generalization can not be expected.

