# OpenReview forum: "Meta-Learning Runge-Kutta"
_ICLR.cc/2020/Conference — Reject_

### Official Review · AnonReviewer2 · 2019-10-21
**Official Blind Review #2**

**Rating:** 3

**Review:**

## Summary ##

The authors present a method for learning a step-size adaptation strategy for Runge-Kutta methods. The principal contributions of the paper are:

1. They define a loss function that better captures global performance of the controller, rather than just local behavior.
2. They propose a set of input features to a step size controller. It includes the intermediate Runge-Kutta evaluation values, which allow the controller to approximate the derivatives of the Jacobian function $g$.
3. They describe a recurrent architecture for adapting step size over time.

Runge-Kutta methods are a workhorse of ordinary differential equations and choosing step size is one of the central challenges involved in their application. Better methods for step size selection would definitely be of broad interest. As the authors point out, existing methods often consist of hand-tuned heuristics---a feature that often suggests machine learning could provide significant improvements.

While the premise of the paper is very promising, I don't think it is ready to be accepted to ICLR at this time. Most significantly, the experimental results are not particularly compelling. I believe the authors should refine their method, aim for better experimental results and resubmit. I have included more specific comments below.

## Specific Comments ##

1. I think the paper would benefit from a clearer description of the RK step-size selection problem. For instance, for a p^th order RK solver, at each time step,

   * Inputs: t, y(t), g  # Also possibly intermediate values from previous time steps.
   * Select a step size h(t)
   * Evaluate g at p different points based on h(t).
   * Use these evaluations to compute a value of y_(t + h_t)

   For those that aren't familiar with these methods (at ICLR there will be many!) I think this would help explain where the authors' method (and the other methods your describe) fits into the larger algorithm.

2. I think a short explanation of error estimation would be helpful in addition to the reference to Butcher. This estimation is critical to step size adaptation. In particular I think this would be clearer if the authors expanded the paragraph at the bottom of page 4, where they describe error in a polynomial in (t_n, h) whose coefficients are derivatives of g.

3.  The authors' proposed loss function (Eq. 5) includes the true value of y at t_n, y(t_n). They acknowledge that this may make it computationally prohibitive, but I think this point warrants further discussion. Does this mean that their loss function can only be used on problems for which we have a closed form solution (such as harmonic oscillators)? I noticed that in their van der Pol experiments, the authors switch to a more standard loss (Eq. 6). Is this because Eq. 5 is intractable in this example? Is there a reasonable approximation to Eq. 5 that could be used when a closed form solution is not known?

4. There are a number of issues with the experiments that I think could use clarification or improvement:

  (a) The authors compare to a 'baseline' but I don't believe this baseline is defined anywhere. Is it one of the adaptation methods described in Section 2?

  (b) In Table 1, the baseline method achieves lower error, while MLRK uses fewer steps. It is difficult to assess if this is an improvement since this this cost-accuracy tradeoff is at the heart of the Lagrangian formulation. Ideally, shouldn't we be able to adjust 'tol' to trace out some Pareto frontier for the cost-accuracy tradeoff? In this case, wouldn't we hope for MLRK to be able to achieve better accuracy given the same computational budget?

  (c) In the van der Pol experiments, the authors switch to local L1 loss (Eq. 6). Was the intention to experiment with local L1 loss and van der Pol provided an interesting class of examples? Or was the reasoning that Eq. 5 is intractable for van der Pol so they had to use local L1 loss? If Eq. 5 can still be evaluated on these experiments, this might be a more convincing comparison.

  (d) The number of steps required was provided for the harmonic oscillator experiments, but not the van der Pol ones. This would be helpful for comparing the methods.

  (e) What is the computational overhead of running an RNN alongside your solver? Although it doesn't tell the whole story, it would be informative to report wall clock time along with the number of steps required for each method.

**Experience Assessment:**

I have read many papers in this area.

**Review Assessment: Checking Correctness Of Derivations And Theory:**

N/A

**Review Assessment: Checking Correctness Of Experiments:**

I assessed the sensibility of the experiments.

**Review Assessment: Thoroughness In Paper Reading:**

I read the paper at least twice and used my best judgement in assessing the paper.

---

> ### Author Response · Authors · 2019-11-09
> **Response to Reviewer 2**
>
> Thank you for your time and feedback.
>
> 1. You addressed your concern that we did not describe the problem of step size control well. We did not consider this as a fundamental preliminary in order to understand the general problem. However, we agree that for the ICLR community may be necessary. We will include a more thorough description.
>
> 2. We think the reference to Butcher is sufficient in the main text, however we agree that an explanation would be helpful and we will add that to the appendix.
>
> 3. In order to use the loss function in Equation 5, we either need a closed form solution as in the experiments with harmonic oscillators or we need to use global error estimation in order to approximate this value. Alternatively, one could consider solving the problem with a very small tolerance parameter to obtain a good approximation of the global error. However, with both these approaches we only obtain an expensive approximation of the global error. In order for our method to work well, we need a global error estimation algorithm that works reliably well, which is still an active area of research. For this reason we left further experiments with the Lagrange loss (Eq. 5) for future work.
>
> 4.
> (a) As you point out, we missed to reference the baseline used in the experiments. It is the one given in Equation (2) and is a standard step size controller used for Runge-Kutta. We will add a comment in the experiment section for the final version of the paper.
>
> (b) This is an interesting point. When we give both methods the same computational butget, they will arrive at different time points in the integration interval and hence, the resulting accuracy of the numerical solution is not comparable. We are uncertain about how to choose a tolerance that allows a clear comparison in both the number of steps and global error. We agree that the results are hard to interpret, however MLRK is within the given tolerance (0.001 * number of steps).
>
> (c) The intention of the experiment with van der Pol equations was that they are an interesting class of ODEs for step size control. As pointed out in 3. we left experiments with global error estimation to future work.
>
> (d) Agreed. We will include this in the final version. Here are the number of steps and computation time for van der Pol equations:
> int |              steps                    |             time                      |
>       | Baseline | Our Method | Baseline | Our Method |
> ---------------------------------------------------------------------------
> 1    | 21.59      | 16.42              | 0.045       | 0.043             |
> 3    | 33.74      | 29.12              | 0.072       | 0.075             |
> 5    | 45.43      | 41.07              | 0.092       | 0.102             |
> 7    | 56.84      | 52.57              | 0.117       | 0.131             |
> 10  | 73.46      | 69.41              | 0.141       | 0.167             |
>
> (e) We agree that the clock times would be informative, here are some values:
> Table for (low-freq) oscillators
> int |              steps                    |             error                     |              time                     |
>       | Baseline | Our Method | Baseline | Our Method | Baseline | Our Method |
> --------------------------------------------------------------------------------------------------------------
> 1    | 3.28         | 3.19               | 0.000006 | 0.000007       | 0.0086     | 0.0074           |
> 3    | 6.82         | 6.04               | 0.000030 | 0.000103       | 0.0144     | 0.0130           |
> 5    | 10.35       | 8.18               | 0.000059 | 0.000326       | 0.0211     | 0.0207           |
> 7    | 13.70       | 10.15             | 0.000089 | 0.000608       | 0.0277     | 0.0293           |
> 10  | 18.85       | 13.03             | 0.000138 | 0.001083       | 0.0407     | 0.0460           |

---

> > ### Comment · AnonReviewer2 · 2019-11-12
> > **Response to Response to Reviewer 2**
> >
> > Thank you for clarifying some of these points. I think adding wall clock times is particularly interesting. I still have a few concerns about the paper:
> >
> > 1. It's unclear whether the Lagrange loss will ever be useful in practice. To apply it, we require a problem that is simple enough for either a closed form solution or a very accurate simulation. In this case, why would we want to use learned step sizes? I think the authors need to make the cases that there could be a family of problems with some "easy" members that could be used to train the model with Lagrange loss and some "hard" members that provide interesting applications. I appreciate that they are leaving a good deal of the work with Lagrange loss for future papers, but I think they need to describe a case where it could be useful or possibly leave it out altogether.
> >
> > 2. To summarize the table in (e), their method yields
> >     * Fewer steps
> >     * Higher error
> >     * Similar wall time
> > Based on this, it's hard to make a case for MLRK.
> >
> > 3. For van der Pol, it's still not totally clear. Figure 3 depicts errors for the baseline, along with three versions of MLRK. It appears that 'partial' and 'grad' perform better than the baseline, but it would be nice to have this data in tabular form, along with timing information for each (it's not clear which method produced the values in the table in (d) above).
> >
> > 4. It seems like the van der Pol oscillators in the training and test set all have $\sigma ~ U(0, 1)$. I'm concerned that this doesn't give sufficient evidence that the model is generalizing. What happens if it is trained on vdP(0, 1) and tested on vdP(1, 2)?
> >
> >
> > Again, I think this is a great premise for a paper, but I don't think the experimental evidence is strong enough at this point. I would still lean toward rejecting the paper, but I think the authors should certainly keep pursuing this idea. In it's current form, I also think this would make a really compelling workshop paper.

---

> > > ### Author Response · Authors · 2019-11-13
> > > **Response to Reviewer 2**
> > >
> > > Thanks for the valuable feedback.
> > >
> > > Concerning (1), while pushing for more general problems is indeed interesting and on our agenda, one of the take-away messages of the present paper is to illustrate that DNNs can actually help speeding up classical initial value problem solvers. Our examples clearly show that classical engineers could benefit from current deep learning. This was also the feedback we got from colleagues from the engineering domain.
> > >
> > > Concerning (2), we disagree in the following sense. Thee errors are indeed higher but within the range of digits where one says that a solution has been found.  So the main point is indeed the “fewer steps” and “similar wall time”. Our goal is to speed up classical engineering techniques without compromising the quality, which our numbers do show.
> > >
> > > Concerning (3):
> > >
> > > int |                      steps                         |                  mean local error                   |                          time                              |
> > >       | baseline| err    | partial| grad  | baseline | err         | partial   | grad   | baseline | err        | partial  | grad  |
> > > ------------------------------------------------------------------------------------------------------------------------------------------------------
> > > 1   | 21.59      | 16.42| 12.40  | 12.09 | 7.17e-4  | 6.58e-4 | 4.01e-4 | 3.74e-4 | 0.0255    | 0.0263 | 0.0254 | 0.0221 |
> > > 3   | 33.74      | 29.12| 25.16  | 24.74 | 6.40e-4  | 5.45e-4 | 2.49e-4 | 2.28e-4 | 0.0405    | 0.0375 | 0.0460 | 0.0403 |
> > > 5   | 45.43      | 41.07| 36.42  | 36.02 | 5.18e-4  | 4.47e-4 | 1.95e-4 | 1.75e-4 | 0.0591    | 0.0517 | 0.0681 | 0.0596 |
> > > 7   | 56.84      | 52.57| 48.34  | 47.97 | 4.97e-4  | 4.16e-4 | 1.57e-4 | 1.39e-4 | 0.0858    | 0.0742 | 0.1036 | 0.0897 |
> > > 10 | 73.46      | 69.41| 65.40  | 65.04 | 4.59e-4  | 3.82e-4 | 1.32e-4 | 1.18e-4 | 0.0971    | 0.0825 | 0.1201 | 0.1065 |
> > >
> > > "err" is slightly faster and uses fewer steps than the baseline while producing smaller local errors during the integration as can be seen in Figure 3 of the paper and in this table. While "partial" and "grad" reduce the number of steps even further, wall time is increased. However one can clearly see that "partial" and "grad“ outperform both the baseline and "err" regarding the local error.
> > >
> > > The method "err" produced the values in the table in (d) of the previous comment.
> > >
> > > Finally about (4),  indeed, this is an interesting setting that we will definitely explore. Thanks for pointing this out. However, even the current results demonstrate already the benefit that learning to learn can have for classical engineering tasks.

---

> > > > ### Comment · AnonReviewer2 · 2019-11-15
> > > > **Thanks for addressing my comments**
> > > >
> > > > The authors have been very responsive to reviewer comments. I think this will make their paper much stronger. In particular, I would strongly suggest that they include the exhaustive table above in the body of the paper.
> > > >
> > > > Regarding (2), the authors make a good point---MLRK stays within the allowed error. I'm still not convinced that "fewer steps" and "similar wall time" constitutes an improvement over the baseline, since wall time is what we really care about. Can the authors make an argument that refinement of their method will lead to a method that is actually faster than the baseline?
> > > >
> > > > As I mentioned above, I think the premise of this paper is really interesting, but I would like to see stronger experimental results. That said, I would be willing to raise my rating to a 5.
> > > >
> > > > Whether the paper is accepted or not, the authors should certainly keep pursuing this line of research.

---

### Official Review · AnonReviewer1 · 2019-10-22
**Official Blind Review #1**

**Rating:** 3

**Review:**

Updated review: Thanks to the authors for their response to my comments. I believe the strong point of this paper is the novel idea, however, I find the justification for that idea incomplete as author's seems to suggest that the proposed method is probably computationally more expensive (which is opposite to the original motivation of the paper).

---------------------------------------------------------------------------------------------------------------------------------------------------------------------

Summary: This paper casts the problem of step-size tuning in the Runge-Kutta method as a meta learning problem. The paper gives a review of the existing approaches to step size control in RK method. Deriving knowledge from these approaches the paper reasons about appropriate features and loss functions to use in the meta learning update. The paper shows that the proposed approach is able to generalize sufficiently enough to obtain better performance than a baseline.

I think this paper, in general, is clear and well-written. I believe the idea of the paper is interesting too.

The paper argues that the main challenge of solving the step size control problem for the RK method is balancing the computation vs accuracy trade-off. Existing methods tackle this problem in different ways and this paper proposes to solve it via meta-learning. However, the paper does not mention how and why meta-learning is expected to tackle this challenge?
So a couple of comments on what set of problems do we expect meta-learning to better tackle this trade-off than the existing methods would have been useful. I am wondering if it is even possible to say something about this in principle?

The paper argues that the idea behind using meta-learning is to learn behaviour from a given class of problems and then generalize to new unseen problems (from the same or different classes).
How do we know that these problems are even from same distributions?
Won't the proposed approach fail spectacularly when the problems are not from the same distribution? It would have been nice if the paper made this distinction even if empirically.

In the experiments section, I could not find/understand what exactly is the baseline the paper is comparing to.

I was more interested in a study that compared the performance of MLRK as the number of instances of the training problems are varied.
This again makes me come back to the original point of computational cost vs accuracy. What is the computational cost of collecting data on 30000 instances of problems? Should we not worry about this cost?
Also, what is the computational cost of the proposed approach and why are we not comparing it to existing approaches/baseline?

minor comments:
what is tol? it tol the same tolerance as lambda.



**Experience Assessment:**

I do not know much about this area.

**Review Assessment: Checking Correctness Of Derivations And Theory:**

I assessed the sensibility of the derivations and theory.

**Review Assessment: Checking Correctness Of Experiments:**

I assessed the sensibility of the experiments.

**Review Assessment: Thoroughness In Paper Reading:**

I read the paper at least twice and used my best judgement in assessing the paper.

---

> ### Author Response · Authors · 2019-11-09
> **Response to Reviewer 1**
>
> Thank you for your time and feedback.
>
> How MLRK tackles the accuracy vs computation challenge: We argue that current hand-designed update rules are constructed based on certain assumptions that - if fulfilled - lead to a minimization of the step control objective. Replacing these hand-designed update rules by a learned one can significantly improve step size control as a learned method aims to minimize the objective without these assumptions.
>
> On what set of problems do we expect meta-learning to better tackle this trade-off: MLRK will lead to improvements whenever an ODE does not satisfy the assumptions made for the corresponding step size algorithm. Examples of ODEs where the assumptions are not met are given in the experiments; the baseline method is not able to control step sizes well for van der Pol equations or double pendulums. This is due to high spikes or chaotic behaviour in the local errors. Other types of ODEs with suddenly changing behaviour in the local errors will likely benefit from MLRK as well.
>
> Next, we want to address the distribution of classes of ODEs. As you point out our method may fail when applied to problems of very different distributions. A controller that is able to generalize to many different classes of problems is the ultimate goal and was proposed as future work in the conclusion. In particular, an approach similar to that of Wichrowska et al. is pointed out as a way to achieve a general step size controller. However, a controller that is specialized to a certain class of problems can also be of great interest for applications that require repeating numerical integrations of ODEs of similar form. For example, if the application continuously needs to integrate some parametric form of ODE with varying parameters our approach can lead to great improvement. We will make sure to point this out in the final version of the paper.
>
> As pointed out, we missed to reference the baseline used in the experiments. It is the one given in Equation (2) and is a standard step size controller used for Runge-Kutta. We will add a comment in the experiment section for the final version of the paper.
>
> Furthermore, you propose an interesting idea to study the effect of a varying number of training instances. We do think this is a compelling idea that deserves further consideration. We are going to design accoring experiments and hope to be able to include them in the final version. Currently, we can not make any comment on the effect of a varying number of training data.
> To address the computational cost of our method, we want to point out that the ODEs of our current experiments are of rather low dimensionality and hence the additional cost of executing an LSTM is comparably high. For higher dimensional problems, the cost of an LSTM is comparably lower and hence we expect significant improvement in the computation cost.
>
> The tolerance parameter tol corresponds to lambda, this is only very briefly touched on in the paragraph on "The Objective of Step Size Control" (Section 2), we will try to make this more clear in the discussion of the performance measures.
>
> Reference:
> Olga Wichrowska, Niru Maheswaranathan, Matthew W. Hoffman, Sergio Gómez Colmenarejo, Misha Denil, Nando de Freitas, and Jascha Sohl-Dickstein. Learned optimizers that scale and generalize. In Proceedings of the 34th International Conference on Machine Learning, pp. 3751–3760, Sydney, Australia, August 2017.

---

### Official Review · AnonReviewer3 · 2019-10-24
**Official Blind Review #3**

**Rating:** 3

**Review:**

The paper proposes to learn the step size for a Runge-Kutta numerical integrator for solving ordinary differential equations initial value problems. The authors frame the stepsize control problem as a learning problem, based on different performance measures, on ODE dependent inputs and on a LSTM for predicting the next step coefficient. Experiments are performed on 3 ODEs by training and testing in different contexts.
The problem of designing adaptive controllers for ODE numerical schemes is interesting and is probably a new application for ML. The paper makes an effort to introduce the necessary background and for reviewing some classical adaptive controller techniques. The description of the method is relatively clear, but could however be largely improved in order to make it more accessible to the audience.  Many of the arguments and proposed ideas come without justification, some definitions should be made more precise. The construction of the training and test sets should be better explained. The experiments show that the proposed approach leads to fewer evaluations but larger mean errors.  The graphics also show that the local error is smaller for the proposed method than for the baselines which is in contradiction with the global error behavior. This should be clarified – the relations between the two error types should be made clear.  The baseline is not defined in the text so that it is difficult to judge the performance. Why not comparing to several adaptive baselines?

In conclusion, this is an interesting topic, the paper proposes new ideas. A more careful writing and especially a better comparison with sota baselines would greatly improve the paper.


------ post rebuttal -----
Thanks for the answers. I still think that the ideas are interesting but that the experiments  do not demonstrate enough of the proposed method. I will keep my score.


**Experience Assessment:**

I have read many papers in this area.

**Review Assessment: Checking Correctness Of Derivations And Theory:**

I assessed the sensibility of the derivations and theory.

**Review Assessment: Checking Correctness Of Experiments:**

I assessed the sensibility of the experiments.

**Review Assessment: Thoroughness In Paper Reading:**

I read the paper at least twice and used my best judgement in assessing the paper.

---

> ### Author Response · Authors · 2019-11-09
> **Response to Reviewer 3**
>
> Thank you for your time and feedback.
>
> You argue that some of our arguments and ideas come without justifications. We motivate different aspects of our method.
> For example, we argue that current hand-designed update rules are constructed based on certain assumptions and that replacing these hand-designed update rules by a learned one can significantly improve step size control.
> Furthermore, we justify the different performance measures and input features. The first performance measure in Equation (5) is based on the fact that in numerical integration we are interested in minimizing both the global error of the approximated solution as well as the computational cost. Our second performance measure in Equation (6) is the underlying objective of most common step size control algorithms and therefore qualifies as an approriate performance measure in our setup as well. The different sets of input features are justified and discussed as well.
> For this reason we are unsure which arguments you are refering to. Can you give a few more specific comments on that regard?
>
> The distribution of training and test data is described in the appendix, for construction of the data an according number of samples were sampled from the distribution. In particular, a parametric form of the ODE is assumed and a distribution over the parameters is defined. We will make sure to include this description in the final version of the paper.
>
> You mention an apparent contradiction in our experiments, however we think this is due to a confusion. The depicted local errors in Figure 3 show the local error of van der Pol equations, whereas the mean global error and number of steps in Table 1 and 2 are evaluated on harmonic oscillators - a different kind of ODE. Furthermore the models in these two experiments are trained with different loss functions. The loss functions and their relation are discussed at length in Section 2. Can you give us feedback if this clarifies your problem? Your comment would help us to decide if we need to clarify this in more detail in the paper.
>
> As you point out, we missed to reference the baseline used in the experiments. It is the one given in Equation (2) and is a standard step size controller used for Runge-Kutta. We will add a comment in the experiment section for the final version of the paper. We agree that a comparison to other step size controllers, e.g. the ones in Equation (3) and (4), is appropriate and try to include them in the experiments of the final version.

---

### Decision · Program_Chairs · 2019-12-19

**Decision:**

Reject

**Comment:**

Summary: This paper casts the problem of step-size tuning in the Runge-Kutta method as a meta learning problem. The paper gives a review of the existing approaches to step size control in RK method. Deriving knowledge from these approaches the paper reasons about appropriate features and loss functions to use in the meta learning update. The paper shows that the proposed approach is able to generalize sufficiently enough to obtain better performance than a baseline.


The paper was lacking in advocates for its merits, and needs better comparisons with other baselines before it is ready to be published.